# The risks of using the chi-square periodogram to estimate the period of biological rhythms

**Michael C. Tackenberg**[1,2], **Jacob J. Hughey**[1,2]*

**1** Department of Biomedical Informatics, Vanderbilt University Medical Center, Nashville, Tennessee, United States of America, **2** Department of Biological Sciences, Vanderbilt University, Nashville, Tennessee, United States of America

* jakejhughey@gmail.com

## Abstract

The chi-square periodogram (CSP), developed over 40 years ago, continues to be one of the most popular methods to estimate the period of circadian (circa 24-h) rhythms. Previous work has indicated the CSP is sometimes less accurate than other methods, but understanding of why and under what conditions remains incomplete. Using simulated rhythmic time-courses, we found that the CSP is prone to underestimating the period in a manner that depends on the true period and the length of the time-course. This underestimation bias is most severe in short time-courses (e.g., 3 days), but is also visible in longer simulated time-courses (e.g., 12 days) and in experimental time-courses of mouse wheel-running and ex vivo bioluminescence. We traced the source of the bias to discontinuities in the periodogram that are related to the number of time-points the CSP uses to calculate the observed variance for a given test period. By revising the calculation to avoid discontinuities, we developed a new version, the greedy CSP, that shows reduced bias and improved accuracy. Nonetheless, even the greedy CSP tended to be less accurate on our simulated time-courses than an alternative method, namely the Lomb-Scargle periodogram. Thus, although our study describes a major improvement to a classic method, it also suggests that users should generally avoid the CSP when estimating the period of biological rhythms.

## Author summary

The chi-square periodogram is a popular method for estimating period length, one of the most important properties of the daily biological rhythms found throughout nature. In this study, we identify a major source of inaccuracy in the chi-square periodogram, and quantify the inaccuracy using a broad array of simulated and experimentally observed biological rhythms. Although we revise the chi-square periodogram calculation to improve its accuracy, we also show that the revised version is still less accurate than an alternative method, the Lomb-Scargle periodogram. Our work thus provides evidence on how to obtain better estimates of the period of biological rhythms.

**Data Availability Statement:** All data, code, and results for this study are available on Figshare (https://doi.org/10.6084/m9.figshare.12805082).

**Funding:** This work was supported by the U.S. National Institutes of Health R35GM124685 to JJH.

The funders had no role in study design, data collection and analysis, decision to publish, or preparation of the manuscript.

**Competing interests:** The authors have declared that no competing interests exist.

## Introduction

Period estimation is critical to the study of circadian rhythms, the endogenous, near-24-h rhythms in physiology and behavior exhibited by organisms from bacteria to humans. For example, the influence of a gene on the circadian clock is assessed in part by estimating the period of rhythms when the gene is absent [1,2]. Effects on period have also been useful for elucidating the contributions of particular cell populations to the central circadian pacemaker in the mammalian brain [3–5] and exploring how the pacemaker reorganizes in response to environmental cues [6,7].

Given a time-course of potentially rhythmic measurements, a standard approach for estimating the underlying period is the chi-square periodogram (CSP) [8]. Previous studies have found that the CSP tends to be less accurate in certain scenarios (e.g., short time-courses) compared to other methods [9–11]. However, these studies did not examine the extent to which the accuracy of each method depends on the combination of time-course length and underlying period, and the reasons for the CSP's lower accuracy remain unclear. Despite its limitations, the CSP remains widely used for estimating period in various types of rhythms across a variety of species [12–16].

Here we used simulated time-courses to uncover and characterize a major source of bias in the standard CSP. The bias results in an underestimation of the period, the degree of which depends on an interaction between the true period and the length of the time-course. Although the bias is most severe in short time-courses (e.g., 3 days), it is observable in simulated time-courses up to at least 12 days. We also see the signature of bias in experimental time-courses of mouse wheel-running and ex vivo bioluminescence. To mitigate the bias, we developed and validated two new versions of the CSP that eliminate the discontinuity from the periodogram and thus estimate period without a period-dependent bias. Although one of these revised versions had greater accuracy and lower variance than the standard CSP, it still tended to be less accurate than alternatives such as the Lomb-Scargle periodogram. Thus, the CSP (in any formulation) should likely not be the first choice for period estimation of biological rhythms.

## Methods

### Data availability

All period estimation methods used in this study are available in an open-source R package called spectr (https://spectr.hugheylab.org). For the Lomb-Scargle periodogram, spectr depends on the lsp function of the lomb R package [9]. For the fast Fourier Transform, spectr depends on the spec.pgram function of the stats package included in R.

### Standard, conservative, and greedy chi-square periodograms

The standard CSP for a time-course of $N$ time-points is based on calculating, for each test period $P$, the ratio of the variance of the means of a series of every $P^{\text{th}}$ time-point ("observed") to the variance of all $K * P$ time-points ("expected"), where $K$ is the number of time-points in each of the $P$ sets of means [8]. This ratio, referred to as $Q_P$, is calculated as:

$$Q_P = \frac{KN \sum_{h=1}^{P} \left( \bar{X}_h - \bar{X} \right)^2}{\sum_{i=1}^{N} \left( X_i - \bar{X} \right)^2}$$

where $X_i$ corresponds to the $i$th time-point, $\bar{X}$ corresponds to the mean of all included time-points (explained below) and $\bar{X}_h$ for a given value of $P$ corresponds to the mean of every $P^{\text{th}}$ time-point.

$K$ is therefore equal to the highest integer quotient of $N / P$:

$$K_{standard} = floor\left(\frac{N}{P}\right)$$

Defined this way, $K$ is subject to change across different test periods and any time-points between $K * P$ and $N$ will be excluded from the calculation for that particular test-period.

In the conservative CSP, $K$ is kept constant across the range of test periods $P_{min}$ to $P_{max}$:

$$K_{conservative} = floor\left(\frac{N}{P_{max}}\right)$$

In this formulation, $K$ remains constant but is completely dependent on $P_{max}$ and the number of omitted timepoints (those between $K * P$ and $N$) is large.

In the greedy CSP, $K$ is allowed to take non-integer values:

$$K_{greedy} = \frac{N}{P}$$

Here the number of time-points omitted remains zero throughout the range of test periods and the value of $K$ is independent of the test period range. In both modified versions, the calculation of the ratio of variances proceeds similarly to the standard CSP. The p-value is based on the approximate chi-square statistic $Q_P$ and the degrees of freedom $P—1$.

## Lomb-Scargle periodogram and fast Fourier transform

We calculated the Lomb-Scargle periodogram using an oversampling factor of 100, and calculated the fast Fourier transform using a zero-padding factor of 100. These factors do not alter spectral resolution (i.e., the ability to distinguish two peaks in the periodogram), but do improve each method's ability to estimate the location of a single peak. The CSP has no equivalent of oversampling or padding [17].

## Simulations and analysis

To simulate time-courses of rhythmic measurements, we used the simphony R package [18]. The simulations varied in true period, amplitude, waveform shape, and time-course length as indicated in the text. Each simulation had a phase sampled uniformly between 0 and the true period.

To simulate non-sinusoidal rhythms, we used a smooth sawtooth wave and a smooth square wave. The smooth sawtooth wave corresponded to:

$$f(t) = \frac{1}{\beta}\sum_{k=1}^{n} sin\left(\frac{2\pi k}{\tau}t\right)\cdot\frac{1}{k^2}$$

where $n = 100$ and $\beta = 1.01495$. The smooth square wave corresponded to:

$$f(t) = \frac{arctan\left(sin\left(\frac{2\pi}{\tau}t\right)\cdot\frac{1}{\delta}\right)}{arctan\left(\frac{1}{\delta}\right)}$$

where $\delta = 0.2$.

Unless otherwise specified, simulations had a sampling interval of 0.1 h and i.i.d. Gaussian noise of mean 0 and standard deviation 1. Simulations with Poisson sampling followed $x(t) = A \cdot f(t) + A + 1$, where $x(t)$ is both the expected value and its variance, $A$ is the amplitude, and $f(t)$ is the rhythmic function that goes between -1 and 1, i.e., sine, smooth sawtooth, or smooth

square. Because a Poisson distribution can only take integer values $\geq 0$ and has variance equal to its mean, it may better approximate activity counts.

For each version of the CSP, we defined the estimated period for a given simulation as the test period with the minimum *p*-value. For the LSP and FFT, we defined the estimated period as the test period with the maximum power. We limited test periods to between 18 and 30 h. We then calculated the error as the difference between the true period and the estimated period.

## Results

### The standard chi-square periodogram shows discontinuities, related to time-course length, that can result in underestimation of the true period

In testing methods of period estimation on simulated time-courses, we found that the chi-square periodogram (CSP) sometimes underestimated true period values > 24 h (Fig 1A), with estimates seemingly fixed near 24 h (Fig 1B). Furthermore, the periodograms showed a discontinuity coinciding with the incorrect period estimate, such that rhythmic power appeared to decrease sharply for candidate periods > 24 h (Fig 1C and 1D). The discontinuity was observable in multiple software implementations of the CSP (S1 Fig), indicating that it was not due to an error in our calculations. In contrast, the Lomb-Scargle periodogram (LSP) showed no such estimation bias (Fig 1A and 1B) or discontinuity (Fig 1E), indicating that the phenomenon was specific to the CSP.

To determine the source of the discontinuity and resulting bias in period estimation, we examined each quantity calculated as part of the CSP (Fig 2). The CSP for a dataset of *N* time-points (Fig 2A) defines a chi-square statistic (sometimes called $Q_P$) for each test period under consideration. The number of time-points in a test period (e.g., 240 for a test period of 24 with time-points every 0.1 h) is referred to as *P*. For each test period, the *N* time-points are arranged in row-major order into a grid with *P* columns and *K* complete rows (Fig 2B, filled squares). The remaining *D* time-points, where $P * K + D = N$, are omitted from the calculation for that test period (Fig 2B, open squares). The ratio of the variance of the column means of the grid and the variance of the non-omitted time-points gives rise to the chi-square statistic and corresponding *p*-value.

Each test period in the CSP therefore has associated values of *K* and *D* (Fig 2C). Although *K* decreases monotonically as *P* (again, the number of time-points comprising a test period) increases, the decrease is not continuous (Fig 2C, middle plot). In our initial simulations, the discontinuity in the periodograms occurred at a test period (24 h) where *K* decreased and *D* increased (Fig 2C).

Because *K* decreases at test periods of which the length of the time-course is a multiple, we simulated time-courses of different lengths to test whether the location of the discontinuity changed accordingly (Fig 3). We first simulated time-courses of 68, 72, and 76 h, in which the true period was 24 h. As expected, the largest discontinuity in the periodograms occurred at test periods of 22.6, 24, and 25.3 h, respectively, all coinciding with changes in *K* (Fig 3A). In the 76-h simulations, an additional discontinuity appeared that also coincided with a change in *K*. In the 68-h simulations, the discontinuity was sufficient to alter the apparent peak of the periodogram. Thus, in these simulations, changing the length of the time-course by 4 h—without changing the underlying waveform—could decrease the estimated period by 1.4 h.

We next simulated time-courses of 72, 144, and 288 h (all multiples of 24 h), again with a true period of 24 h. In each case, the largest discontinuity occurred at a test period of 24 h and all discontinuities coincided with changes in *K* (Fig 3B). Taken together, these results indicate that discontinuities are inherent to the calculation of the standard CSP and lead to the bias in period estimation.

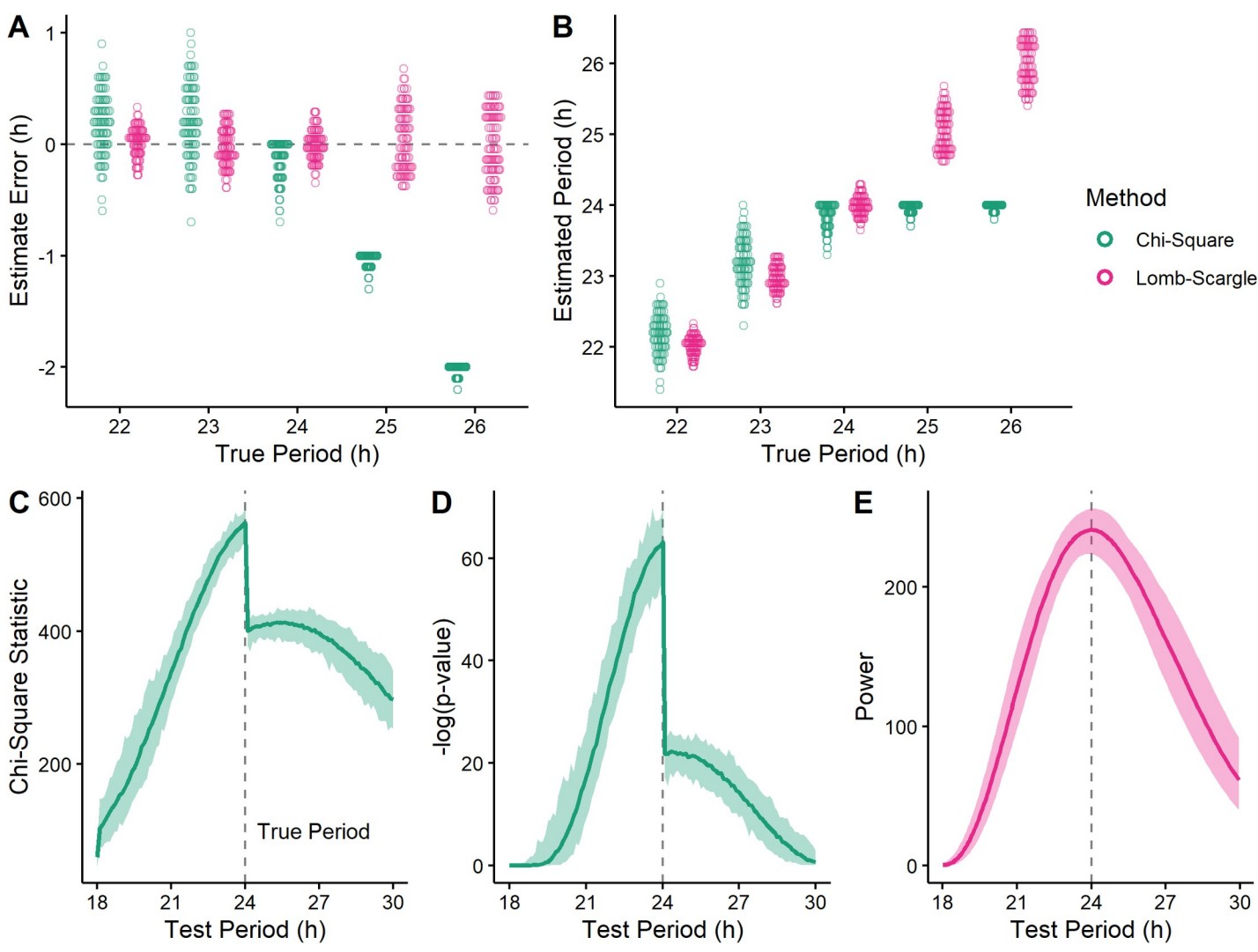

**Fig 1. Period underestimation by the chi-square periodogram (CSP) coincides with a discontinuity in the periodogram.** (A) Period estimation error and (B) estimated period for the CSP and Lomb-Scargle periodogram (LSP) on simulated time-courses that had various values of true period. Each point represents a simulated time-course (100 for each true period). Each time-course had a length of 3 days and a sinusoidal rhythm with amplitude 2. The dotted line in (A) indicates an error of 0. From the simulated time-courses with true period 24, periodograms of the (C) chi-square statistic and (D) corresponding -log(*p*-value) for the CSP, and (E) rhythmic power for the LSP. Each darker curve represents the median, each lighter shaded region represents the range.

To further characterize the bias, we applied the CSP to simulated time-courses that varied in length as well as in period and amplitude of the underlying rhythm. We found that as the length of the time-course increased, the magnitude of the bias decreased and the range of true periods affected by the bias narrowed (Fig 3C). For example, in 6-day time-courses the bias affected true periods between approximately 24 and 24.8 h, whereas in 9-day time-courses the bias affected true periods between approximately 24 and 24.4 h. In these ranges of true periods, the CSP's discontinuity caused the peak of the periodogram to always occur at a test period of 24 h. At the right-most edge of each range, a small change in true period led to a large change in estimated period. In 12-day time-courses the bias affected two distinct ranges of true periods, each range corresponding to a discontinuity. Importantly however, both the magnitude of the bias and the ranges of affected true periods were independent of rhythm amplitude

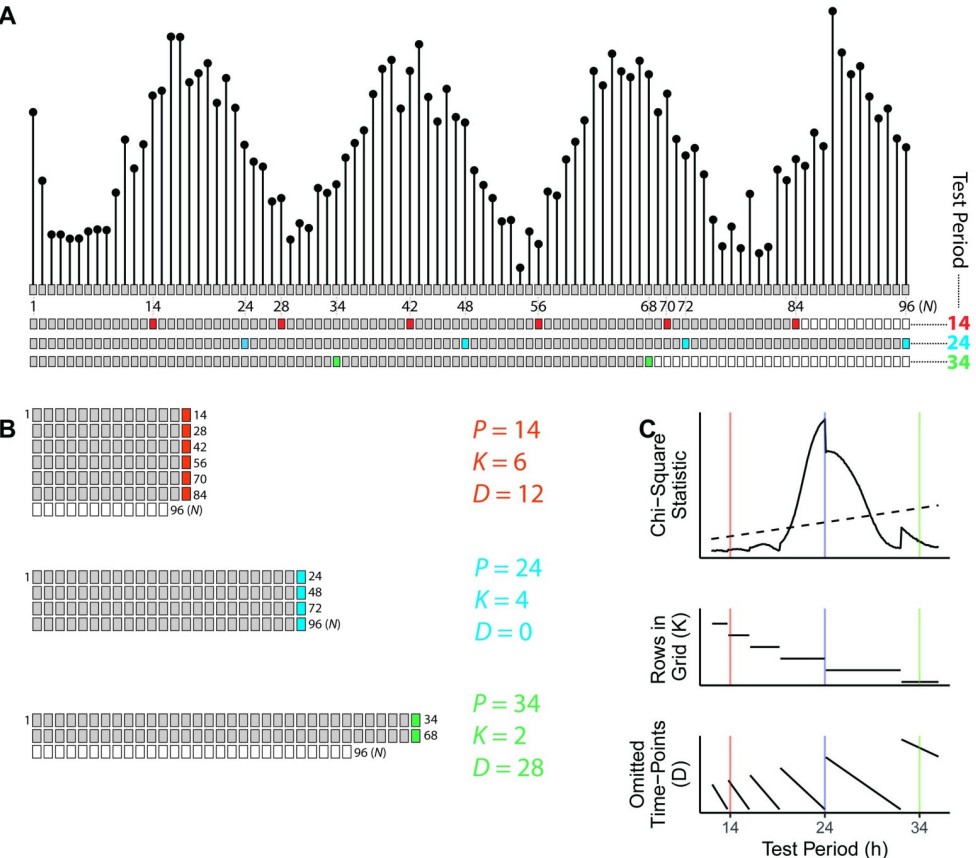

**Fig 2. Schematic and example calculation of the chi-square periodogram.** (A) Top, a simulated time-course of length 96 h, having a sinusoidal rhythm with amplitude 4 and true period 24 h. Each dot and box represent a time-point, with 1 h between time-points. Bottom, sequential layout of time-points for three test periods, where every *P*th time-point is colored according to the test period. Other time-points are grey if used in the calculation, and white if omitted. (B) Time-points organized into a grid for each test period, each with particular values of *K* and *D*. The variance of means from each complete row is used to calculate the chi-square statistic. (C) Periodogram of chi-square statistics and associated *K* and *D* values for the simulated time-course. The colored vertical lines represent the three test periods. The dashed line in the periodogram indicates the traditional alpha = 0.05 significance threshold.

(S2 Fig). Thus, although the CSP bias diminishes with longer time-courses, it does not diminish with higher-amplitude rhythms.

## Revising the calculation of the chi-square periodogram removes the discontinuity and reduces the bias

We next sought to modify the calculation of the CSP to avoid discontinuities and hopefully remove the estimation bias. We developed two new versions of the CSP: the conservative CSP and greedy CSP. In the conservative CSP, *K* is held constant across all test periods at a value equal to the maximum number of complete rows possible at the highest test period considered (i.e., *K* in the conservative CSP corresponds to the minimum *K* in the standard CSP; Fig 4A). In the greedy CSP, the columns of the *P*-by-*K* grid are allowed to have unequal numbers of rows, and *K* is set to the mean number of rows per column (which might not be an integer; Fig 4B). Compared to the standard CSP, the conservative version tends to discard more data, whereas the greedy version never discards any. In both cases, the calculation of variances proceeds similarly to the standard CSP and so the periodograms maintain their relationship to the

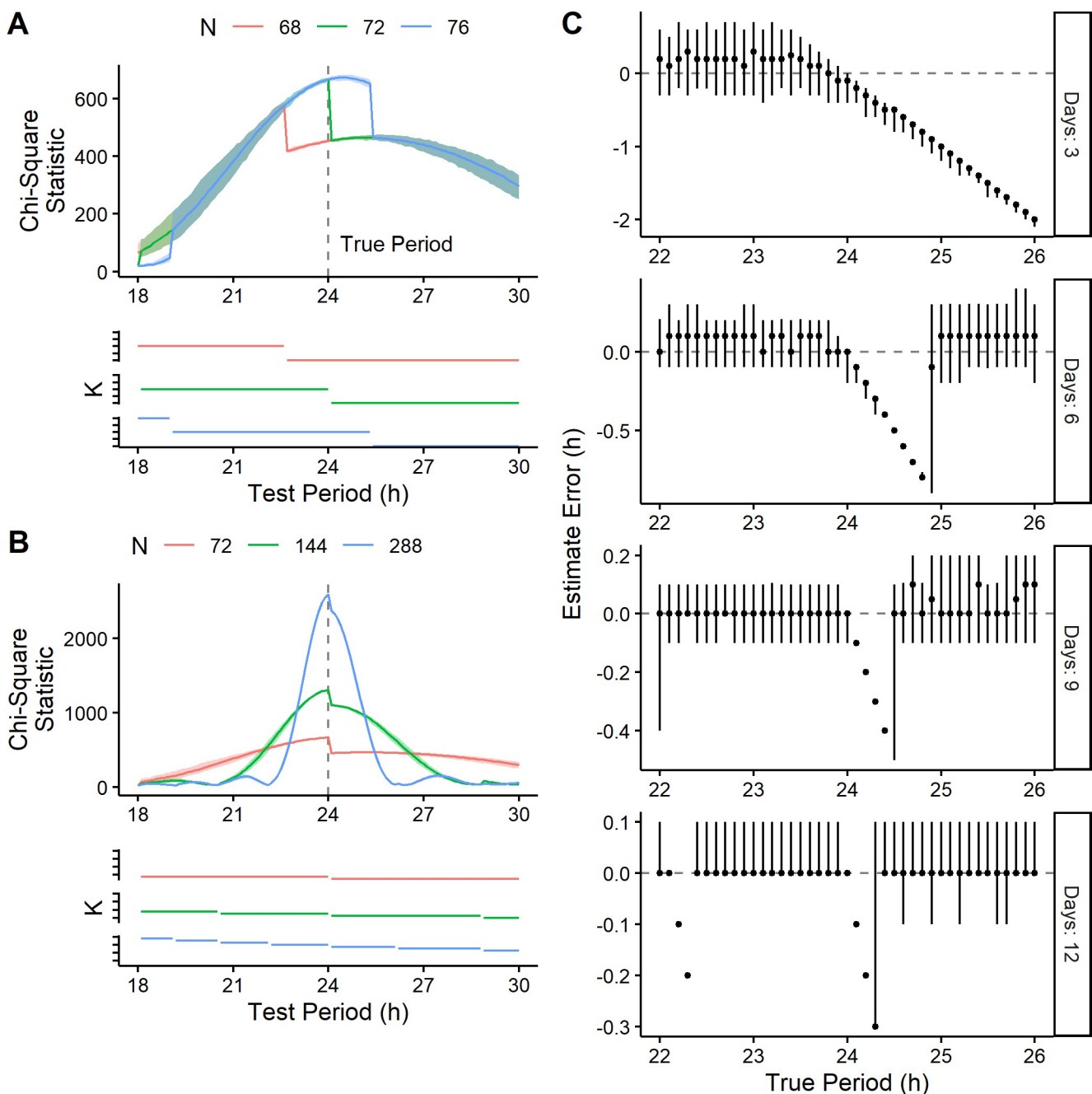

**Fig 3. Time-course length influences the location of discontinuities in the CSP and the resulting period estimation bias.** Periodograms and the associated *K* values from simulated time-courses of length (A) 68, 72, and 76 h and (B) 72, 144, and 288 h (100 time-courses for each length). Each time-course had a sinusoidal rhythm with a true period of 24 h and an amplitude of 2. In the periodograms, each darker curve represents the median, each shaded region represents the range. (C) Period estimate error from simulated time-courses of various lengths and with various values of true period. Each point represents the median of 100 time-courses (all with a sinusoidal rhythm with amplitude 2), and each vertical line represents the 5th-95th percentile range.

chi-square distribution. The periodograms calculated using the new CSP versions show no discontinuities (Fig 4A and 4B).

To determine whether these modifications reduced the bias of the CSP, we tested each version on a comprehensive set of simulations (see Methods). As baseline methods for comparison, we used the LSP and a padded FFT. For each simulation and each method, we calculated

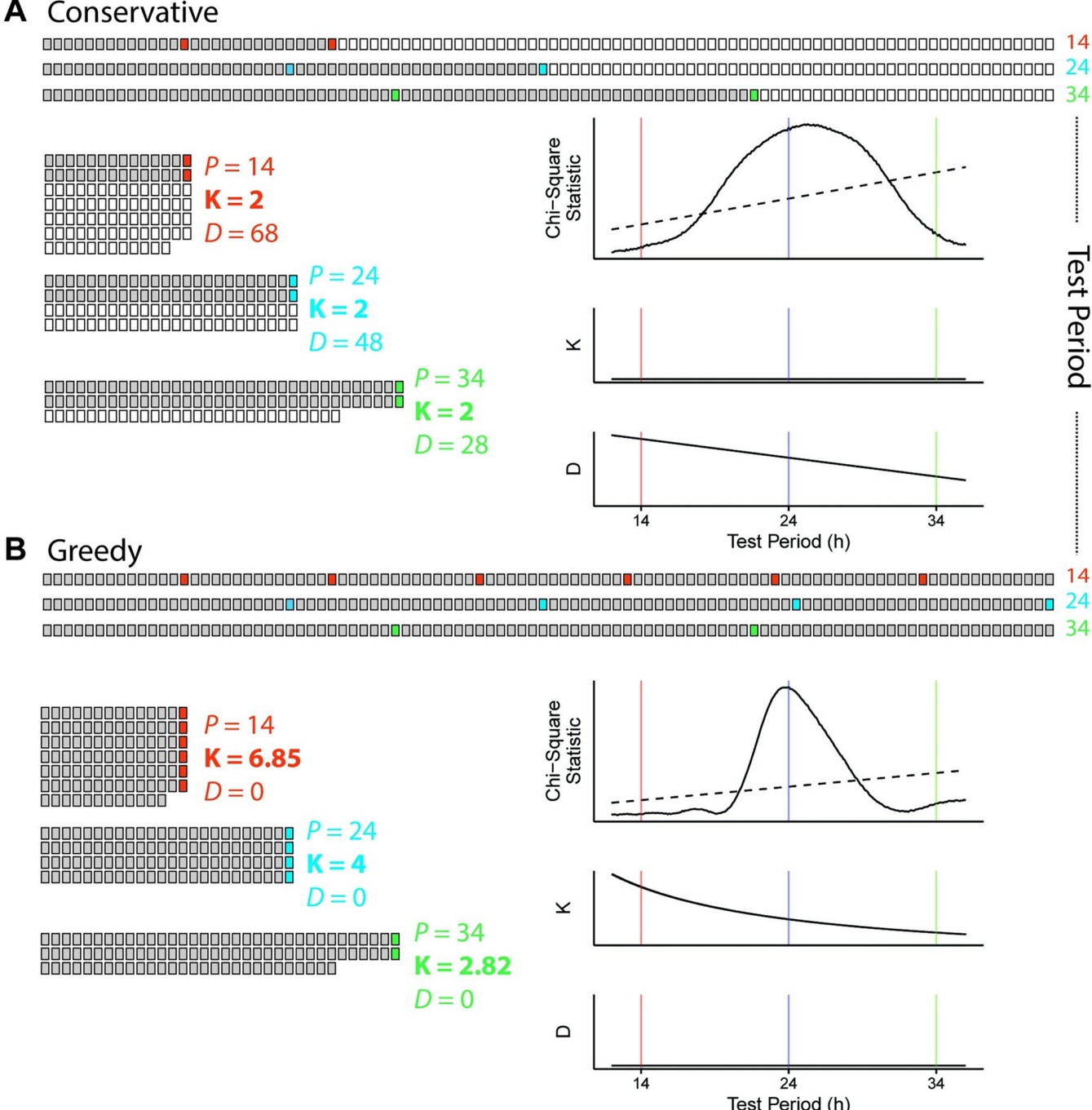

**Fig 4.** Removing the discontinuity of the chi-square periodogram by redefining K. Schematic and example calculation for the (A) conservative CSP and (B) greedy CSP. In (A) and (B), top represents time-points laid out sequentially, left indicates time-points organized into a grid for three different test periods, and right indicates the associated chi-square statistics and values of *K* and *D*. Grey or colored boxes represent time-points used in the calculation, white boxes indicate omitted time-points. The example calculations use the same simulated time-course as in Fig 2.

the estimate error and the absolute estimate error. Compared to the distributions of estimate error from the standard CSP, those from the conservative and greedy CSPs were centered closer to zero and less dependent on the true period (Figs 5A and S3–S5), indicating that the

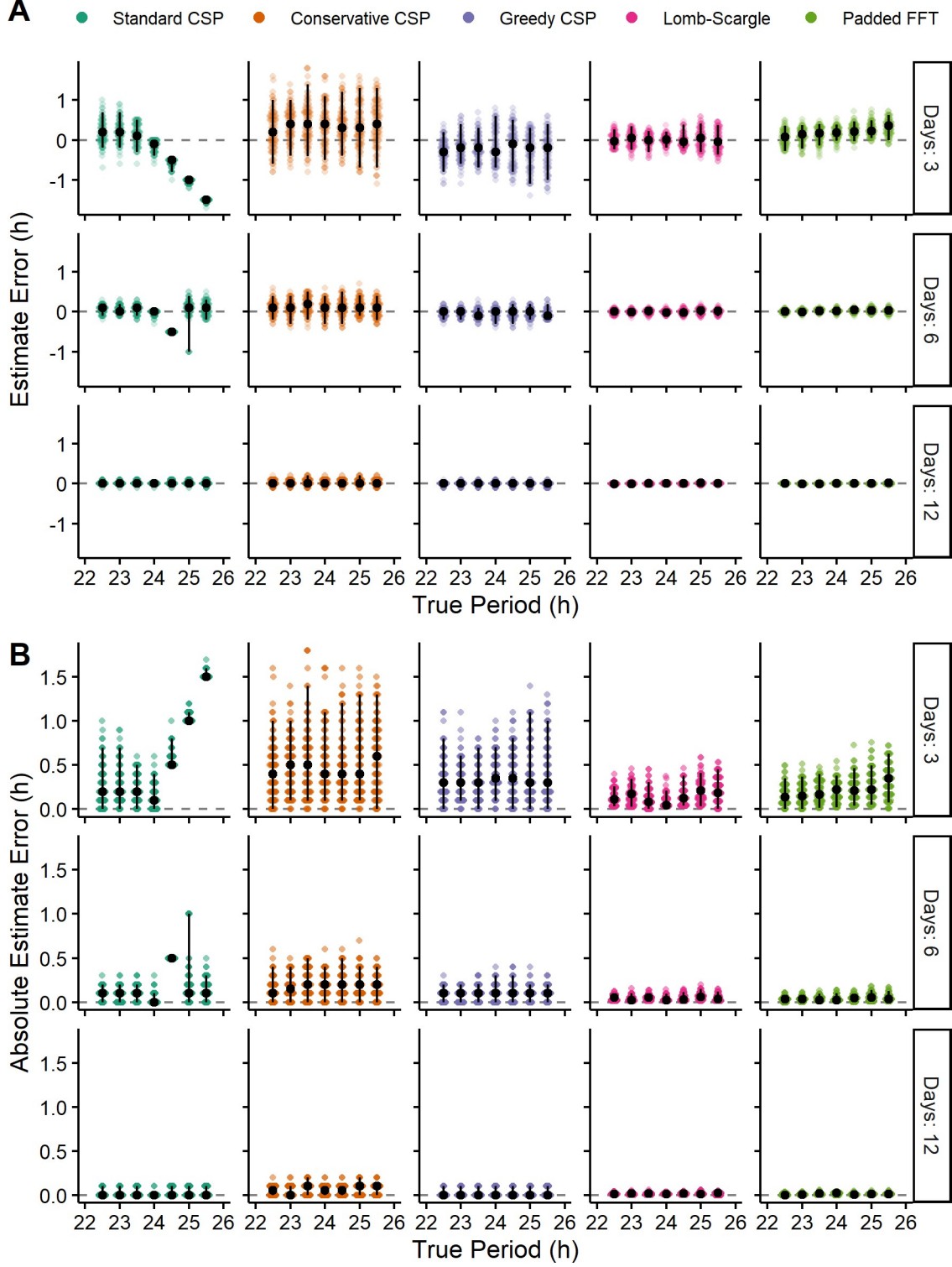

**Fig 5. Revised versions of the CSP have reduced bias, but still tend to have lower accuracy than the LSP and padded FFT.** (A) Estimate error and (B) absolute estimate error for various methods on simulated time-courses of various lengths and with various values of true period. Each point represents a simulated time-course, with 100 time-courses per combination of length and true period. Each time-course had a sinusoidal rhythm with amplitude 2. Black circles and vertical black lines represent the median and 5th-95th percentile range, respectively.

new CSP versions do indeed have reduced bias. Compared to the conservative CSP, the greedy CSP tended to give distributions of estimate error with lower variance (S1 Table), consistent with the latter tending to omit fewer time-points from the calculation. The lower variance of the greedy CSP translated to a lower absolute estimate error, whereas the higher variance of the conservative CSP caused its absolute estimate error to often be higher than that of the standard CSP (Fig 5B and S2 and S3 Tables). For example, in 3-day simulations with a period of 23 h (where the discontinuity has little effect), the standard, conservative, and greedy CSPs had mean absolute errors of 0.295, 0.505, and 0.337 h, respectively. In 3-day simulations with a period of 25 h, however, where the discontinuity strongly affects the standard CSP, the mean absolute errors of the three methods were 1.042, 0.553, and 0.417 h, respectively. These trends were independent of the rhythm's waveform and amplitude (S3 and S4 Figs and S2 and S3 Tables) and were also present in time-courses of lower temporal resolution (S5 Fig and S4 Table). These results suggest that of the three CSP versions, the greedy CSP strikes the best balance between bias and variance so as to achieve accurate period estimation.

However, even the greedy CSP tended to have higher variance in estimate error and higher absolute estimate error compared to the LSP and the padded FFT (Figs 5 and S3–S5 and S2–S4 Tables). These patterns persisted in simulations with Poisson noise rather than Gaussian noise (S6 and S7 Figs, S5 and S6 Tables). Thus, although our revisions to the CSP have improved its accuracy, they have not improved it to the level of alternative methods for estimating rhythmic period.

## The bias of the standard chi-square periodogram is visible when applied to experimental data

To complement our analyses of simulated time-courses, we next estimated the free-running period in time-courses of mouse wheel-running activity [7]. We truncated the wheel-running data to six different lengths: 72, 120, and 168 h (3, 5, and 7 24-h days) and 69, 115, and 161 h (3, 5, and 7 23-h days). Because the true period in each truncated time-course is unknown, we compared the estimates of the standard CSP against those of the LSP (Fig 6A). Similarly to the standard CSP's estimates on simulated time-courses, its estimates here tended to accumulate at values of which the time-course length is a multiple. This tendency became stronger in shorter time-courses. Furthermore, the periodograms for the standard CSP showed discontinuities at the expected locations based on time-course length (Fig 6B), whereas the periodograms for the LSP showed no discontinuities (Fig 6C). We observed similar trends when we analyzed time-courses of bioluminescence from ex vivo slices of mouse suprachiasmatic nuclei (SCN) expressing PER2::LUCIFERASE [7] (S8 Fig). These results suggest that the bias of the standard CSP is reproducible in experimental data.

## Discussion

The chi-square periodogram remains widely used in studies of circadian rhythms despite a variety of alternatives [11,17]. Previous work on the CSP suggested that, for time-courses shorter than 5–10 days, the test statistics ($Q_P$) may not follow a chi-square distribution and thus the associated $p$-values may be misleading [8,19]. While these previous findings highlight the risk of overinterpreting a given test period's power, our current findings highlight a potentially more severe risk: systematically misestimating the test period with the highest power. Although the risk we identified lessens in longer time-courses, our simulations indicate that it is largely independent of the rhythm's shape and amplitude. Thus, even a highly "statistically significant" rhythm in a moderate-length circadian time-course is vulnerable.

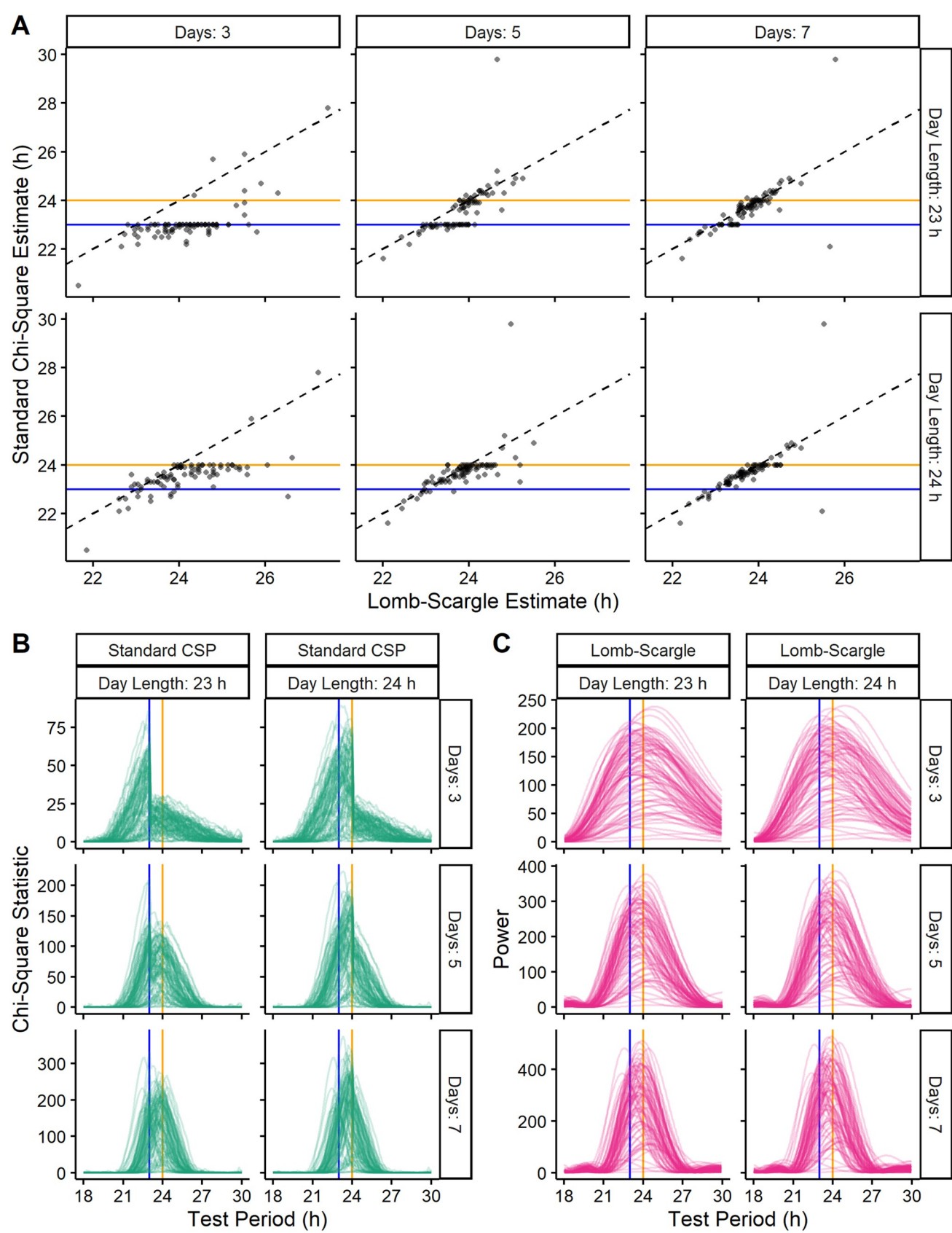

**Fig 6. The bias and underlying discontinuity of the standard CSP are present in the analysis of experimental data.** (A) Scatterplots of estimated period for the LSP and standard CSP on time-courses of mouse wheel-running truncated to various lengths based on various numbers of days and day lengths. Periodograms for the (B) standard CSP and (C) LSP on the same truncated time-courses. Blue and orange lines indicate 23 and 24 h, respectively.

The advantage of simulated time-courses is that they allow precise quantification of how different properties of a rhythm affect the accuracy of each method. The disadvantage is that their similarity to experimental time-courses is only approximate. For example, each of our simulations had a single, stationary period. This simplification seems reasonable, however, because stationarity is a primary assumption of every method we evaluated. Furthermore, methods to detect changes in a circadian rhythm over time require longer time-courses than we considered (e.g., 20 days) [20]. Overall, the consistency between our analyses of simulated and experimental time-courses supports the robustness of our findings.

Three factors make typical circadian time-courses especially vulnerable to the bias of the standard CSP. First, circadian experiments tend to be time-, labor-, and cost-intensive, which creates pressure to use the shortest time-course possible. Second, circadian time-courses are often arbitrarily collected in multiples of 24 h. Third, free-running circadian periods tend to be near (but not exactly) 24 h. Thus, based on our findings, the standard CSP will sometimes mis-estimate the period as exactly 24 h, particularly if the true period is slightly longer than 24 h.

The scale of such misestimations is not trivial. We observed period estimation errors of up to 0.5 h, 0.4 h, and 0.3 h in 6-day, 9-day, and 12-day simulated time-courses, respectively. For comparison, the standard deviation of the free-running period of wild-type C57 mice is only 0.06 h [21], and well-known clock gene mutations can alter period by 0.5–1.5 h [1]. Moreover, because the magnitude of the error depends on the true period, the standard CSP could also misestimate the difference in period between conditions. For instance, in 6-day time-courses in which one condition has a true period of 23.9 h and another condition 24.4 h, using the standard CSP could lead to an estimated period difference of only 0.1 h. Conversely, if the two conditions have true periods of 24.4 and 24.6 h, the estimated period difference could be 0.6 h. Importantly, these errors in estimated period and period difference do not diminish with larger sample sizes.

To minimize such errors, we developed two alternative approaches to calculate the chi-square periodogram. The first approach, which we call the conservative CSP, was useful to confirm the discontinuity as the source of bias but is impractical for two reasons. First, the large amount of omitted data produces period estimates with a large variance, and second, the chi-square statistic for each test period depends on the largest test period under consideration. The second approach, the greedy CSP, shows greatly reduced bias while maintaining low variance compared to the standard CSP. Nonetheless, the greedy CSP tends to be less accurate than the Lomb-Scargle periodogram (which can also accommodate unevenly spaced time-points) and the padded fast Fourier transform. Based on previous work [11], these two methods are likely not alone in outperforming the greedy CSP. The standard and modified versions of the CSP, as well as the LSP and padded FFT, are included in a new R package called spectr.

In conclusion, the risks of using the chi-square periodogram seem to outweigh any benefits. Overall, our findings support the following, in decreasing order of preference: (1) using alternative methods instead of any version of the CSP, (2) using the greedy CSP instead of the standard CSP, and (3) inspecting the periodograms of the standard CSP for discontinuities. In this way, our findings could strengthen the contribution of accurate period estimation to the study of biological rhythms.

## Supporting information

**S1 Fig. The discontinuity in the chi-square periodogram is visible in multiple software implementations. (A)** A simulated time-course with a sinusoidal rhythm of amplitude 2. **(B)** The corresponding chi-square periodogram calculated by the xsp R package. (C) ClockLab actogram view of the pre-loaded "Sample 1" dataset from days 4 to 6 and **(D)** the corresponding chi-square periodogram.
(TIF)

**S2 Fig. Period estimation bias in the standard CSP is independent of rhythm amplitude.** Estimate error on simulated time-courses of various lengths having a sinusoidal rhythm with various values of amplitude and true period. Each point represents the median of 100 time-courses, and each vertical line represents the 5th-95th percentile range.
(TIF)

**S3 Fig. Bias and variance of each method are largely independent of waveform shape.** Waveforms of **(A)** sinusoidal, **(C)** smooth square, and **(E)** smooth sawtooth rhythms of amplitude 2. Black curves indicate expected rhythm, grey regions indicate one standard deviation of the Gaussian noise above and below. Estimate error for each method on simulated time-courses of various lengths and having a **(B)** sinusoidal, **(D)** smooth square, or **(F)** smooth sawtooth rhythm. Each point represents a simulated time-course, with 100 time-courses per combination of length and true period. Black circles and vertical black lines represent the median and 5th-95th percentile range, respectively.
(TIF)

**S4 Fig. Relative bias and variance of each method are largely independent of rhythm amplitude.** Sinusoidal rhythms of amplitude **(A)** 1, **(C)** 2, and **(E)** and 4. Black curves indicate expected rhythm, grey regions indicate one standard deviation of the Gaussian noise above and below. Estimate error for each method on simulated time-courses of various lengths and having a rhythm with amplitude **(B)** 1, **(D)** 2, or **(E)** 4. Each point represents a simulated time-course, with 100 time-courses per combination of length and true period. Black circles and vertical black lines represent the median and 5th-95th percentile range, respectively.
(TIF)

**S5 Fig. Relative bias and variance of each method in time-courses of lower temporal resolution (sampling interval of 20 minutes instead of 6 minutes). (A)** Estimate error and **(B)** absolute estimate error for various methods on simulated time-courses of various lengths and with various values of true period. Each point represents a simulated time-course, with 100 time-courses per combination of length and true period. Each time-course had a sinusoidal rhythm with amplitude 2. Black circles and vertical black lines represent the median and 5th-95th percentile range, respectively.
(TIF)

**S6 Fig. Bias and variance of each method are largely independent of waveform shape in simulated time-courses with measurements sampled from a Poisson distribution.** Waveforms of **(A)** sinusoidal, **(C)** smooth square, and **(E)** smooth sawtooth rhythms of amplitude 2. Black curves indicate expected rhythm, grey regions indicate one standard deviation above and below. Estimate error for each method on simulated time-courses of various lengths and having a **(B)** sinusoidal, **(D)** smooth square, or **(F)** smooth sawtooth rhythm. Each point represents a simulated time-course, with 100 time-courses per combination of length and true period. Black circles and vertical black lines represent the median and 5th-95th percentile range, respectively.
(TIF)

**S7 Fig. Relative bias and variance of each method are largely independent of rhythm amplitude in simulated time-courses with measurements sampled from a Poisson distribution.** Sinusoidal rhythms of amplitude **(A)** 1, **(C)** 2, and **(E)** and 4. Black curves indicate expected rhythm, grey regions indicate one standard deviation above and below. Estimate error for each method on simulated time-courses of various lengths and having a rhythm with amplitude **(B)** 1, **(D)** 2, or **(E)** 4. Each point represents a simulated time-course, with 100 time-courses per combination of length and true period. Black circles and vertical black lines represent the median and 5th-95th percentile range, respectively.
(TIF)

**S8 Fig. The bias and underlying discontinuity of the standard CSP are present in the analysis of PER2::LUCIFERASE SCN recordings. (A)** Scatterplots of estimated period for the LSP and standard CSP on time-courses truncated to various lengths based on various numbers of days and day lengths. Periodograms for the **(B)** standard CSP and **(C)** LSP on the same truncated time-courses. Blue and orange lines indicate 23 and 24 h, respectively.
(TIF)

**S1 Table. Mean and standard deviation of estimate error, as well as the mean of the absolute estimate error, for each combination of method, true period length, signal amplitude, and waveform shape shown in Fig 5.**
(TXT)

**S2 Table. Mean and standard deviation of estimate error, as well as the mean of the absolute estimate error, for each combination of method, true period length, signal amplitude, and waveform shape shown in S3 Fig.**
(TXT)

**S3 Table. Mean and standard deviation of estimate error, as well as the mean of the absolute estimate error, for each combination of method, true period length, signal amplitude, and waveform shape shown in S4 Fig.**
(TXT)

**S4 Table. Mean and standard deviation of estimate error, as well as the mean of the absolute estimate error, for each combination of method, true period length, signal amplitude, and waveform shape shown in S5 Fig.**
(TXT)

**S5 Table. Mean and standard deviation of estimate error, as well as the mean of the absolute estimate error, for each combination of method, true period length, signal amplitude, and waveform shape for simulations with Poisson-distributed noise shown in S6 Fig.**
(TXT)

**S6 Table. Mean and standard deviation of estimate error, as well as the mean of the absolute estimate error, for each combination of method, true period length, signal amplitude, and waveform shape for simulations with Poisson-distributed noise shown in S7 Fig.**
(TXT)

## Acknowledgments

We thank Allison Leich-Hilbun for input on the revised calculations for the chi-square periodogram. We thank Josh Schoenbachler for helping to develop unit tests for the spectr package. We thank Jeff Jones and Doug McMahon for helpful comments on the manuscript.

## Author Contributions

**Conceptualization:** Michael C. Tackenberg, Jacob J. Hughey.

**Data curation:** Michael C. Tackenberg, Jacob J. Hughey.

**Formal analysis:** Michael C. Tackenberg, Jacob J. Hughey.

**Funding acquisition:** Jacob J. Hughey.

**Investigation:** Michael C. Tackenberg, Jacob J. Hughey.

**Methodology:** Michael C. Tackenberg, Jacob J. Hughey.

**Project administration:** Jacob J. Hughey.

**Resources:** Michael C. Tackenberg, Jacob J. Hughey.

**Software:** Michael C. Tackenberg, Jacob J. Hughey.

**Supervision:** Jacob J. Hughey.

**Validation:** Michael C. Tackenberg, Jacob J. Hughey.

**Visualization:** Michael C. Tackenberg, Jacob J. Hughey.

**Writing – original draft:** Michael C. Tackenberg, Jacob J. Hughey.

**Writing – review & editing:** Michael C. Tackenberg, Jacob J. Hughey.

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
