## [Decision Letter · Decision Letter 0]

27 Nov 2020

Dear Dr. Hughey,

We are pleased to inform you that your manuscript 'The risks of using the chi-square periodogram to estimate the period of biological rhythms' has been provisionally accepted for publication in PLOS Computational Biology.

Best regards,

Ulrik R. Beierholm

Associate Editor

PLOS Computational Biology

Kim Blackwell

Deputy Editor

PLOS Computational Biology

Reviewer's Responses to Questions

**Comments to the Authors:**

Reviewer #1: This paper has been revised to address the issues raised by all reviewers. The inclusion of single-cell analysis is most welcome and adds significantly to the utility of the authors' approach. I have no additional comments.

Reviewer #2: I am happy with the revisions made.

**Have all data underlying the figures and results presented in the manuscript been provided?**

Reviewer #1: Yes

Reviewer #2: Yes

PLOS authors have the option to publish the peer review history of their article (what does this mean?). If published, this will include your full peer review and any attached files.

Reviewer #1: No

Reviewer #2: No

---

## [Editor Report · Acceptance letter]

30 Dec 2020

PCOMPBIOL-D-20-01910 

The risks of using the chi-square periodogram to estimate the period of biological rhythms

Dear Dr Hughey,

I am pleased to inform you that your manuscript has been formally accepted for publication in PLOS Computational Biology. Your manuscript is now with our production department and you will be notified of the publication date in due course.

With kind regards,

Livia Horvath
